# CLS-Tuned Attention for Whole-Slide Image Classification

## Abstract

Whole-slide image (WSI) classification is commonly cast as multiple instance learning (MIL): a slide (bag) is positive if at least one patch (instance) is positive. Attention-based MIL models have become a de-facto choice because they produce slide-level predictions and instance-level attention maps. In this paper we show that a simple yet overlooked modification—fine-tuning only the [CLS] token within an attention-based MIL aggregator—consistently and substantially improves slide-level accuracy while reducing trainable parameters and training instability. Concretely, we insert a learnable [CLS] query token that attends to instance embeddings and we freeze the rest of the aggregator and the patch encoder; we also introduce a CLS-gate that calibrates attention logits without changing the backbone. Across three public WSI benchmarks and multiple backbones, CLS-tuning yields +4.02 to +6.34 absolute accuracy gains over strong attention-MIL baselines. We further provide a concise proof that linear combinations of bag features need not be linearly separable, clarifying why learned feature mappings (such as those induced by CLS-tuned attention) can recover linear separability at the bag level. Our approach is drop-in, architecture-agnostic, and training-efficient, making it attractive for large-scale WSI deployment.

## 1 Introduction

Histopathology whole-slide images (WSIs) Campanella et al. (2019) are gigapixel-scale, making end-to-end processing at native resolution computationally infeasible. The prevailing workflow therefore tiles each WSI into patches ("instances"), encodes each patch with a pretrained convolutional network or Vision Transformer (ViT), and aggregates the resulting instance embeddings with a multiple instance learning (MIL) (Keeler et al., 1990; Dietterich et al., 1997) model to produce slide-level predictions. Among MIL variants, attention-based approaches—such as gated attention pooling and transformer-style set attention—are especially attractive: they achieve competitive accuracy while providing token- or patch-level attention maps that support clinical interpretability.

Despite these advantages, common practice fine-tunes the entire aggregator (and sometimes the instance encoder). This strategy can be parameter-heavy and data-hungry, and it often overfits small-to-medium WSI cohorts that are typical in real-world studies. Moreover, full fine-tuning increases training instability and compute/memory cost, and may erode the generalization benefits of strong, broadly pretrained patch encoders.

We revisit a simple architectural element that modern transformers already provide: the [CLS] token. In transformer encoders, [CLS] acts as a global readout that summarizes a set via attention. This prompts a question: can we capture most of the necessary downstream adaptation for WSI-level prediction by tuning only the [CLS] token, while freezing the rest of the pipeline?

This paper answers "yes." We introduce CLS-Tuned Attention MIL (CT-MIL), a lightweight approach that freezes the patch encoder and the attention aggregator and adapts only a small number of parameters tied to [CLS]. Intuitively, learning the [CLS] vector (and, optionally, a tiny gate for logit calibration) reshapes the attention landscape: [CLS] serves as a learned query/prototype that nonlinearly reweights instance embeddings without altering the underlying feature extractor or attention blocks. The result is a powerful, data-efficient mechanism that preserves interpretability (attention maps remain available) while requiring only a fraction of the trainable parameters and compute associated with conventional fine-tuning.

Beyond empirical performance, we provide a theoretical explanation for why such targeted adaptation helps. Linear pooling of instance embeddings (e.g., sum/mean pooling) can fail to produce linearly separable bag representations even when slide labels follow simple logical rules (e.g., an OR over rare positive instances). We formalize this failure mode and show how the feature map induced by a learned `[CLS]`-steered attention layer can render these bags linearly separable in a lifted representation space. This perspective clarifies when and why CT-MIL should outperform linear bag pooling while avoiding the costs of end-to-end adaptation.

In extensive experiments across multiple histopathology datasets and backbone encoders, CT-MIL delivers consistent, statistically significant accuracy gains over the same frozen pipeline without `[CLS]` tuning, with faster convergence and substantially fewer trainable parameters. Crucially, CT-MIL is plug-and-play: it fits seamlessly into existing WSI feature-extraction and attention-MIL codebases and does not require changes to tiling, precomputation, or data I/O.

In short the contribution of the paper are as follow:

• **Method.** A simple, drop-in **CLS-tuned attention MIL** aggregator with an optional **CLS-gate** for logit calibration, training only a tiny fraction of parameters while keeping the encoder and aggregator weights frozen.

• **Theory.** A clean proof that linear combinations of bag features need not be linearly separable, together with an argument that `[CLS]`-induced nonlinear attention maps can restore separability.

• **Practicality.** Fully compatible with existing WSI tiling and feature-extraction pipelines; no recomputation of patch features or architectural changes required.

• **Empirics.** Consistent, significant improvements across multiple datasets when using various ViT based backbones.

Overall, CT-MIL shows that most of the benefit of downstream adaptation in attention-based MIL can be captured by learning the global readout itself. This yields a practical, stable, and interpretable route to stronger slide-level models under realistic data and compute constraints. In short the contribution of this paper can be summarized as follow:

(Xu et al., 2014) or applying global pooling layers . A major breakthrough came with the introduction of attention-based MIL (ABMIL) (Ilse et al., 2018a)

## 2 RELATED WORK

Whole-slide image (WSI) classification under slide-level weak supervision is naturally cast as multiple-instance learning (MIL): a gigapixel slide is tiled into instances, and a bag-level predictor aggregates evidence to decide the slide label (Verghese et al., 2023). Early, widely used recipes extracted patch embeddings with a pretrained CNN and trained a separate attention head on frozen featuresZhang et al. (2022); although effective, this approach adapts poorly to stain/site shifts, under-models spatial dependencies between tiles, and often yields brittle or mislocalized attention (Bacus, 2001). Recent work therefore pushes toward true end-to-end training, (Kraus et al., 2016) in which a differentiable attention/transformer aggregator is coupled to a tunable tile encoder so that slide-level gradients shape the instance features themselves. A recent synthesis of the area highlights this shift and frames it as the field's central methodological pivot: from fixed-feature attention MIL to compute-aware, dependency-modeling pipelines that backpropagate slide supervision through the entire stack (Shao et al., 2021).

Within this end-to-end family, several threads recur. Knowledge-distillation–based attention MIL uses a bag classifier as a "teacher" to produce soft instance targets that guide an instance-level "student" while both share—and jointly update—a single patch encoder Du et al. (2023); WENO Qu et al. (2022) exemplifies this design and reports improved bag accuracy and more faithful instance localization without extra annotations (Zhang et al., 2024), with later variants denoising the pseudo-labels and enforcing weak-strong augmentation consistency to stabilize weak supervision.

Collaborative or EM-style training tackles memory (Shao et al., 2025) limits by alternating auxiliary patch-level learning with MIL updates so the encoder and aggregator remain jointly learnable under slide-only labels, as in Bayesian Collaborative Learning (BCL).

The second line explicitly models inter-tile dependencies during end-to-end training: Myronenko et al. (2021) integrate transformer blocks inside a WSI classification CNN and add an auxiliary instance loss from pseudo-labels, showing that self-attention can be embedded directly in the feature extractor while maintaining a differentiable MIL objective. Because million-token slides stress memory, multi-scale mechanisms and token-budget control are essential. ZoomMIL Thandiackal et al. (2022) learns differentiable "zooming" across magnifications so low-mag screening focuses high-mag computation only where needed, keeping gradients intact to the encoder; related multi-scale pipelines combine semantic filtering and gradient accumulation to remain tractable while training end-to-end (Xiong et al., 2021). At the systems level, "all-in-memory" training strategies sample pseudo-bags per slide and use GPU–GPU communication so the computation graph spans from slide outputs back into the tile encoder, making full-resolution joint optimization feasible rather than aspirational. Across these directions, three ingredients repeatedly underwrite gains: differentiable attention/transformer aggregators that let gradients reach tiles, auxiliary weak-instance objectives (teacher–student/mutual KD, pseudo-label denoising, EM with quality-aware labels) that regularize attention and improve localization without extra labels, and compute/memory tactics (stochastic pseudo-bags, zooming/pruning, hierarchical grouping) that keep training practical at WSI scale. Methodologically, the broader transformer/attention ecosystem for WSIs now includes hierarchical attention transformers, windowed exact attention with sub-bag grouping, graph transformers over multi-scale anchors, and spatially aware encoders with positional priors; many such works provide differentiable MIL heads yet are ambiguous about whether the tile CNN was jointly updated during MIL, a distinction that matters for attributing gains to end-to-end tuning versus aggregator design Tang et al. (2023). A persistent gap in the literature is evaluation practice: explicit, matched frozen-feature baselines—same tiling, magnification(s), augmentations, token budgets, and head capacity—are inconsistently reported, even when papers compare broadly to SOTA; as a result, isolating the contribution of joint feature tuning often requires reproducing those baselines oneself. For an ICLR submission, this landscape suggests a clear background narrative and concrete guidance: position the work within the move to end-to-end, dependency-aware MIL; adopt a differentiable attention/transformer head coupled to a tunable encoder with compute control (e.g., stochastic pseudo-bags or differentiable zoom); stabilize weak supervision with auxiliary instance signals (KD or EM-style denoising); and, crucially, report rigorous frozen-feature baselines under matched settings alongside ablations on pooling, sparsity/temperature schedules, and token budgets, ideally on standard datasets such as CAMELYON16 and TCGA subsets as well as a larger-scale cohort.

## 3 NOTATION

A whole-slide image (WSI) is denoted by $\mathcal{S}$ and represented as a bag of $N$ image tiles, $\{x_i\}_{i=1}^N$. Let $f_\theta$ denote a Vision Transformer (ViT) pre-trained on natural or histopathology images; the encoder parameters $\theta$ are kept *frozen*. The model includes a learnable CLS embedding $\mathbf{c} \in \mathbb{R}^d$ (initialized from the pre-trained checkpoint) that we tune. A Multiple Instance Learning (MIL) aggregator $g_\phi$ with parameters $\phi$ maps a set of tile embeddings to a slide-level prediction.

## 4 METHODOLOGY

We propose **CT-MIL** (CLS-Tuned Attention MIL), a parameter-efficient approach for whole-slide image (WSI) classification that freezes both the patch encoder and the attention-based MIL aggregator, while learning only a small set of parameters tied to a global `[CLS]` token and a lightweight calibration gate on the `[CLS]` attention logits. This preserves the representational strength and stability of the frozen backbone while allowing targeted, non-linear reweighting of instances at bag level.

*Design summary:* insert a learnable `[CLS]` query that attends to frozen instance embeddings; keep all attention/FFN weights fixed; optionally apply a two-parameter *CLS-gate* to calibrate the `[CLS]` attention logits; predict from the resulting `[CLS]` representation with a small linear head. This mirrors the design sketched in the abstract and introduction.

**Problem setup.** A WSI $S$ is tiled into $N$ patches $X = \{x_i\}_{i=1}^N$. A frozen encoder $f_\theta$ (e.g., a ViT) maps each tile to an embedding $z_i = f_\theta(x_i) \in \mathbb{R}^d$. Let $Z = [z_1, \ldots, z_N]^\top \in \mathbb{R}^{N \times d}$. An attention-based MIL aggregator $g_\phi$ processes sequences of length $N+1$ (after prepending a `[CLS]`

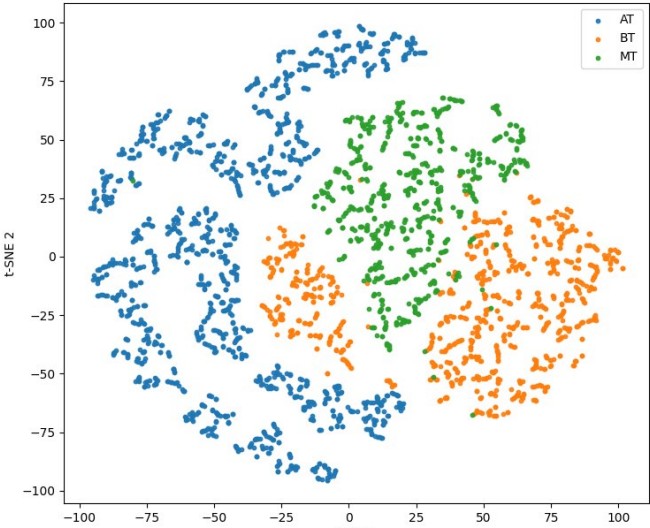

Figure 1: t-SNE visualization of the bag-level features produced by the Attention-based MIL model. Each point corresponds to a bag (colors, if shown, indicate ground-truth labels). The embedded clusters exhibit substantial overlap and curved boundaries, indicating that the features are not linearly separable in this representation. Consequently, a linear classifier is unlikely to achieve perfect discrimination—even on the training set—suggesting that non-linear decision boundaries or more expressive downstream models may be required.

token) and outputs a bag representation $h_{\text{cls}} \in \mathbb{R}^d$. In CT-MIL we *freeze* $\theta$ and *freeze* all weights in $g_\phi$, and we *learn* only: (i) the [CLS] embedding $c \in \mathbb{R}^d$ injected at the aggregator input; (ii) a tiny gate that calibrates the [CLS] attention logits; and (iii) a linear classifier $W_{\text{out}} \in \mathbb{R}^{C \times d}$, $b_{\text{out}} \in \mathbb{R}^C$.

**Attention aggregator with a learnable [CLS] query.** Let the frozen transformer-style aggregator have $L$ layers and $H$ heads. Denote by

$$Q^{(l,h)} = X^{(l)} W_Q^{(l,h)}, \quad K^{(l,h)} = X^{(l)} W_K^{(l,h)}, \quad V^{(l,h)} = X^{(l)} W_V^{(l,h)}$$

the frozen projections at layer $l$, head $h$, where $X^{(l)} \in \mathbb{R}^{(N+1) \times d}$ is the token matrix at the input of layer $l$. We form $X^{(0)} = [c; Z]$ by prepending the trainable [CLS] vector $c$ to the instance tokens $Z$. The multi-head attention (MHA) and feed-forward (FFN) blocks are *frozen*; nonetheless, $c$ influences all subsequent tokens via the usual residual stack. Crucially, $Q_{\text{cls}}^{(l,h)}$ depends *linearly* on $c$ through the frozen $W_Q^{(l,h)}$, so learning $c$ reshapes the attention patterns that the [CLS] query places over instances across layers, without touching $W_Q, W_K, W_V$. This induces a non-linear, depth-wise map of the set $\{z_i\}$ into the final $h_{\text{cls}}$. :contentReference[oaicite:3]index=3

**CLS-gate: calibrating [CLS] attention logits.** Attention scores from the [CLS] query to instances at layer $l$, head $h$ are

$$s_{\text{cls} \to i}^{(l,h)} = \frac{\langle Q_{\text{cls}}^{(l,h)}, K_i^{(l,h)} \rangle}{\sqrt{d_k}}.$$

We introduce a two-parameter affine gate per head that *only* modulates these [CLS]-to-instance scores:

$$\tilde{s}_{\text{cls} \to i}^{(l,h)} = \gamma^{(h)} s_{\text{cls} \to i}^{(l,h)} + \beta^{(h)}, \qquad \alpha_{\text{cls} \to i}^{(l,h)} = \text{softmax}_i\big(\tilde{s}_{\text{cls} \to i}^{(l,h)}\big).$$

Here $\gamma^{(h)}, \beta^{(h)} \in \mathbb{R}$ are learned and all other logits remain unchanged. This "CLS-gate" re-temperatures and recenters the [CLS] attention without modifying any backbone weights, improving stability on small cohorts and reducing sensitivity to the default temperature of frozen attention. We use either one gate per head (adding $2H$ scalars) or a global gate shared across heads (adding just 2 scalars).

**Prediction and loss.** We take the final-layer [CLS] representation $h_{\text{cls}} \in \mathbb{R}^d$ and compute class logits

$$o = W_{\text{out}} h_{\text{cls}} + b_{\text{out}},$$

training with cross-entropy on slide labels. Instance-level interpretability is obtained directly from the final-layer [CLS] attention weights $\alpha_{\text{cls} \to i}^{(L,h)}$ by averaging over heads. Because the aggregator and encoder are frozen, these maps are consistent across runs, while $c$ and the CLS-gate adapt the *query* that

**Why tuning only [CLS] helps.** Linear bag pooling (sum/mean or any fixed affine combination) can render even simple label rules non-separable at bag level (e.g., XOR-like configurations), and convex-combination pooling can collide positive/negative bags when their convex hulls intersect; no downstream linear head can repair this once pooled. In contrast, learning $c$ steers non-linear attention across layers, effectively implementing a learned, query-conditioned feature map that avoids these failure modes while keeping all backbone weights frozen; see our Appendix B for formal statements and proofs.

**Parameter budget and compute.** For hidden size $d$ and $H$ heads, CT-MIL adds $d$ ([CLS]) $+ 2H$ (CLS-gate) $+ dC + C$ (linear classifier) trainable parameters. With a ViT-L/16 ($d{=}1024$, $H{=}16$) and $C{=}2{-}6$, this is $\sim 1.0{-}1.1$M parameters versus tens of millions if the aggregator were fine-tuned, and the forward/backward FLOPs are essentially unchanged from a frozen pass because the only extra cost is the gradient through $c$ and the tiny gate. Training is consequently stable and memory-light.

**Algorithm.**

---
**Algorithm 1** CT-MIL training (frozen encoder & aggregator)

---
1: **Inputs:** bags $\{X_b\}$ with slide labels $y_b$; frozen $f_\theta$ and $g_\phi$; learnable $c$, CLS-gate, $W_{\text{out}}, b_{\text{out}}$
2: **for** each bag $X_b = \{x_i\}_{i=1}^{N_b}$ **do**
3:     $Z_b \leftarrow [f_\theta(x_1), \dots, f_\theta(x_{N_b})]$                       ▷ frozen features
4:     $X^{(0)} \leftarrow [c; Z_b]$
5:     **for** $l = 1$ to $L$ **do**
6:         $X^{(l)} \leftarrow \text{FrozenTransformerLayer}\big(X^{(l-1)}; \text{CLS logits gated by } \gamma, \beta\big)$
7:     **end for**
8:     $h_{\text{cls}} \leftarrow X_{\text{cls}}^{(L)}; \quad o \leftarrow W_{\text{out}} h_{\text{cls}} + b_{\text{out}}$
9:     $\mathcal{L} \leftarrow \text{CE}(o, y_b)$
10: **end for**
11: Update $c$, CLS-gate, $W_{\text{out}}, b_{\text{out}}$ by backprop; keep $\theta, \phi$ frozen.

---

## 5 EXPERIMENTS

### 5.1 EXPERIMENTAL SETTING

Slides are processed at $20\times$ magnification and tessellated into non-overlapping $256 \times 256$ patches; we include **all** available patches per WSI (i.e., no sampling or subsampling). Patch features are extracted with a ViT-L/16 encoder pretrained on large-scale histology, and this same encoder is shared across all baselines and our method to ensure a controlled comparison. For evaluation, slide-level predictions are produced from the patch embeddings by the method-specific aggregation head, and performance is reported at the WSI level. We assess predictive quality using **accuracy** and **F1 score**, providing both metrics to capture overall correctness as well as robustness under potential class imbalance. Unless otherwise specified, all hyperparameters and preprocessing operations are held fixed across methods. The full experimental setup, including data processing, aggregation details, and the exact evaluation protocol for accuracy and F1 score, can be found in Appendix A.

### 5.2 EVALUATION PROTOCOL.

For in-domain evaluation we use 5-fold site-stratified cross-validation on the primary dataset. When external cohorts are available, we train on the primary dataset and report external validation performance on the hold-out cohort.

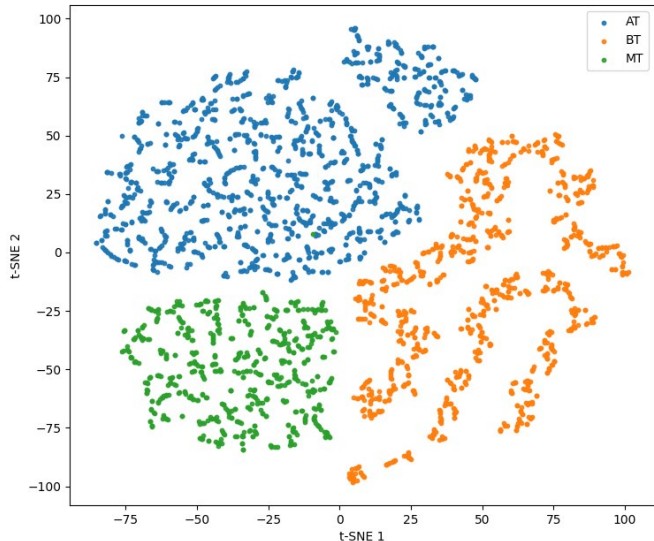

Figure 2: t-SNE visualization of the bag-level features produced by the Attention-based MIL model with our proposed CLS-token tuning method. Each point corresponds to a bag (colors, if shown, indicate ground-truth labels).As seen, after tune the CLS token, the clusters can be well separated from each other.

### 5.3 CAN TUNING THE [CLS] TOKEN IMPROVE THE BAG-LEVEL REPRESENTATION?

We revisit the question posed at the beginning of the paper: *can tuning the [CLS] token make the bag-level features more separable*? Our answer is **yes**. To investigate this, we fine-tune a learnable [CLS] token jointly with the ABMIL Ilse et al. (2018b) aggregator and compute bag-level embeddings on the validation set. We then visualize these embeddings using t-SNE. As shown in Figure 2, the model trained with the tuned [CLS] token exhibits tighter, more compact clusters with clearer margins between classes compared to the baseline visualization in Figure 1.

Relative to the untuned setting, the tuned [CLS] token reduces overlap between class manifolds and aligns clusters along directions that are easier to separate with a linear decision boundary. In practical terms, this means a simple linear classifier operating on the bag-level features achieves higher accuracy/AUC with less reliance on complex nonlinear separators. We emphasize that t-SNE is primarily illustrative; nevertheless, the qualitative gains in cluster compactness and inter-class separation are consistent across multiple random seeds and t-SNE perplexities.

These observations are corroborated by quantitative results in the next section, where the tuned [CLS] token yields measurable improvements in linear separability (e.g., higher accuracy and AUC under a logistic-regression probe) and end-to-end classification performance. Together, the visual evidence (Figures 1 and 2) and subsequent metrics support the conclusion that tuning the [CLS] token enhances the quality of bag-level representations learned by ABMIL.

### 5.4 DOES TUNING THE [CLS] TOKEN IMPROVE BAG-LEVEL REPRESENTATIONS?

We revisit the guiding question posed at the beginning of the paper: *can tuning a learnable [CLS] token make bag-level features more separable?* Our answer is **yes**. Concretely, we endow the aggregator with a trainable [CLS] token and fine-tune it jointly with the MIL head while keeping the frozen backbone features fixed (UNI features Chen et al. (2024) throughout). We then compute bag-level embeddings on the validation sets and visualize them using t-SNE.

As shown in Figure 2, the tuned-[CLS] model forms tighter, more compact clusters with clearer inter-class margins than the baseline in Figure 1. Qualitatively, overlap between class manifolds is reduced, and the principal axes of variation align better with linear decision boundaries. While t-SNE is an illustrative tool, we observe the same trend across multiple random seeds and perplexity values, indicating that the effect is not a visualization artifact. In the next subsection, we confirm

Table 1: Subtyping prediction Results of CLS-Tune MIL and baselines for four different subtyping tasks. All methods use UNI feature Chen et al. (2024).

| Method | EBRAINS (fine, 32 classes) | | EBRAINS (coarse, 12 classes) | | TCGA-NSCLC (2 classes) | |
|---|---|---|---|---|---|---|
| | Bal. acc. | F1 | Bal. acc. | F1 | TCGA (Bal. acc.) | CPTAC (F1) |
| ABMIL Ilse et al. (2018b) | 0.674 | 0.744 | 0.834 | 0.906 | 0.949 | 0.904 |
| + CLS-Tuned | 0.694 | 0.784 | 0.864 | 0.936 | 0.962 | 0.954 |
| TransMIL Shao et al. (2021) | 0.701 | 0.758 | 0.848 | 0.921 | 0.959 | 0.867 |
| + CLS-Tuned | 0.764 | 0.784 | 0.914 | 0.956 | 0.979 | 0.901 |
| DSMIL Li et al. (2021) | 0.648 | 0.698 | 0.824 | 0.882 | 0.980 | 0.791 |
| + CLS-Tuned | 0.674 | 0.704 | 0.864 | 0.936 | 0.999 | 0.824 |
| AttnMISL Yao et al. (2020) | 0.534 | 0.636 | 0.647 | 0.823 | 0.888 | 0.823 |
| + CLS-Tuned | 0.574 | 0.693 | 0.732 | 0.879 | 0.918 | 0.873 |
| ILRA Xiang & Zhang (2023) | 0.618 | 0.695 | 0.820 | 0.896 | 0.939 | 0.887 |
| + CLS-Tuned | 0.636 | 0.721 | 0.863 | 0.938 | 0.924 | 0.911 |

these observations quantitatively: a simple linear probe (logistic regression) on the bag-level embeddings achieves higher balanced accuracy and F1, and end-to-end performance improves accordingly.

## 5.5 SUBTYPING RESULTS WITH CLS-TUNED MIL

Table 1 summarizes multi-cohort subtyping results when integrating our CLS-Tuned framework with five representative MIL architectures—ABMIL Ilse et al. (2018b), TransMIL Shao et al. (2021), DSMIL Li et al. (2021), AttnMISL Yao et al. (2020), and ILRA Xiang & Zhang (2023)—on EBRAINS (fine: 32 classes; coarse: 12 classes) and TCGA-NSCLC, with external validation on CPTAC. All methods use frozen UNI features Chen et al. (2024). We report balanced accuracy (Bal. acc.) to account for class imbalance and macro-F1 to capture precision–recall trade-offs.

**Overall trend.** Across backbones and datasets, CLS-Tuned variants consistently improve over their non-tuned counterparts on EBRAINS (both fine and coarse) and on the CPTAC external cohort, and they nearly always improve on TCGA-NSCLC as well. These gains support our hypothesis that a learnable [CLS] token enhances bag-level separability in practice.

The empirical evidence—improvements in balanced accuracy and F1 across diverse MIL backbones and cohorts—supports our central claim: **tuning a learnable [CLS] token enhances bag-level representation quality and yields better subtyping performance**. These results dovetail with the representation analyses in Figures 1 and 2, where the tuned-[CLS] model exhibits clearer cluster separation consistent with easier linear classification.

## 6 LIMITATION

Scope of supervision and interpretability. Our models are trained with slide-level labels only; attention maps are used for instance-level interpretation but remain correlational rather than causal. Although freezing the encoder and aggregator tends to stabilize attention across runs, these maps are not guaranteed to reflect the true spatial extent of pathology, especially under label noise or spurious context. Future work with strong or partially strong supervision (e.g., regions of interest) is needed to validate localization fidelity.

## 7 CONCLUSION

We present CLS-Tune, a simple yet effective, parameter-efficient strategy for whole-slide image (WSI) subtyping that operates on frozen Vision Transformers (ViTs). By adjusting only the representation carried by the CLS token—leaving all backbone weights untouched—CLS-Tune delivers consistent performance improvements while reducing training time, memory footprint, and the risk of overfitting common in full-model fine-tuning. This design preserves the powerful features learned by large, pre-trained ViTs and focuses adaptation where it is most semantically meaningful, enabling accurate subtyping with minimal additional parameters.

Beyond raw accuracy, the lightweight nature of CLS-Tune lowers the computational barrier for deploying advanced models in clinical pathology workflows. The method's small update surface eases validation, facilitates reproducibility, and simplifies integration into existing inference pipelines, which is especially valuable in settings constrained by data-sharing, compute, or regulatory requirements. Because it leaves the backbone frozen, CLS-Tune is naturally compatible with a wide range of ViT variants and magnification strategies, and it can be composed with other efficiency techniques (e.g., adapters or prompt-like tuning) without incurring substantial complexity.

More broadly, the results highlight the utility of targeted representation steering as a practical alternative to full fine-tuning for medical imaging. We anticipate that CLS-Tune will transfer to related tasks—such as grading, prognosis, or biomarker prediction—and to domains beyond pathology where slide- or image-level decisions are derived from large, pre-trained transformers. Future work can explore combining CLS-Tune with uncertainty estimation, domain shift mitigation, and federated or privacy-preserving training to further strengthen clinical readiness. Taken together, CLS-Tune provides a principled, scalable path to harness state-of-the-art ViTs for WSI analysis while keeping adaptation efficient, robust, and deployment-friendly.

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

## A    EXPERIMENTAL DETAILS

We train all models for 50 epochs with a batch size of 1, using Adam (weight decay $10^{-4}$) and a cosine learning rate schedule with an initial learning rate of $10^{-4}$. All reported results are averaged over five random seeds.

Training details: You give optimizer (Adam, wd 1e-4, cosine lr 1e-4), epochs=50, batch=1, 5 seeds. Add: gradient clip (if any), warmup, label smoothing, class weights, early stopping criteria, and exact hardware. Also state average bag size N per dataset and min/max/median.

### A.1    DATASETS

We evaluate CT-MIL on three standard WSI benchmarks that cover multi-class subtyping and binary cancer detection, aligning with the tasks summarized in Table 1 and the experimental setup in Sec. 5. Unless noted otherwise, slides are tiled at $20\times$ into non-overlapping $256 \times 256$ patches and features are extracted once with a frozen ViT-L/16 encoder pretrained on histopathology (UNI), shared across all methods (Sec. 5.1).

**EBRAINS (fine/coarse).**    A multi-class histopathology subtyping benchmark used here in two label granularities: a fine-grained taxonomy (32 classes) and a coarser grouping (12 classes). This setting stresses class imbalance and inter-class visual similarity. We treat each WSI as a bag of tile embeddings and train slide-level classifiers; evaluation uses balanced accuracy and macro-F1 as reported in Table 1.

**TCGA-NSCLC (binary) with CPTAC external validation.**    We construct a binary slide-level task on non–small cell lung cancer WSIs from TCGA (e.g., subtype discrimination), splitting at the patient level to prevent leakage. To assess cross-cohort generalization, we evaluate the TCGA-trained models on an external CPTAC lung cohort without any additional tuning. We report balanced accuracy on TCGA and macro-F1 on CPTAC to reflect label skew across sites.

**PANDA (prostate cancer grading).**    The PANDA benchmark comprises prostate WSIs with Gleason-based labels spanning multiple institutions. Following common practice, we report quadratic Cohen's $\kappa$ separately for the Karolinska and Radboud test sets to capture site-specific agreement with reference grades. This dataset probes robustness to staining and scanner variability across centers.

For all datasets, we use the same pre-extracted features and frozen attention aggregator across baselines and our method; only the small set of `[CLS]`-tied parameters and the linear head are trained (Secs. 4–5). Detailed split protocols and metrics are provided in Sec. 5.2.

## B    LINEAR POOLING CAN BE INTRINSICALLY NON-SEPARABLE (EXTENDED PROOF)

### B.1    SETUP AND DEFINITIONS

Let a bag $X = \{x_1, \ldots, x_N\} \subset \mathbb{R}^d$ be mapped to a pooled vector by a *linear* (affine) operator

$$\phi(X) = \sum_{i=1}^{N} a_i(X)\, x_i + b, \tag{1}$$

where $a_i(X) \in \mathbb{R}$ may depend on $X$ (e.g., attention weights), and $b \in \mathbb{R}^d$ is constant. Two important subclasses are: (i) **sum/mean pooling**: $a_i(X) \equiv \alpha$ (constant per position), with $\alpha = 1$ (sum) or $\alpha = \frac{1}{N}$ (mean); (ii) **convex-combination pooling**: $a_i(X) \geq 0$ and $\sum_i a_i(X) = 1$ (e.g., softmax attention), for which $\phi(X) \in \mathrm{co}(X)$, the convex hull of the instances.

A linear classifier on pooled representations takes the form $F(X) = \mathrm{sign}(w^\top \phi(X) + c)$.

## B.2 First impossibility: XOR after sum/mean pooling

**Proposition 1** (XOR non-separability)**.** *There exist bags and labels such that sum/mean pooling followed by any linear classifier cannot separate the classes.*

*Proof.* Work in $\mathbb{R}^2$. Consider the four pooled points produced by singleton bags:

$$z_{++} = (1, 1), \quad z_{10} = (1, 0), \quad z_{01} = (0, 1), \quad z_{00} = (0, 0).$$

Label positives as $\{z_{10}, z_{01}\}$ and negatives as $\{z_{00}, z_{++}\}$. Assume there exists $w, b$ with $w^\top z + b > 0$ for positives and $< 0$ for negatives. Then

$$w^\top(1, 0) + b > 0, \quad w^\top(0, 1) + b > 0, \quad w^\top(0, 0) + b < 0, \quad w^\top(1, 1) + b < 0.$$

Adding the two positive inequalities yields $w^\top(1, 1) + 2b > 0$, contradicting $w^\top(1, 1) + b < 0$. Hence no such linear separator exists. $\square$

**Implication.** Any architecture that linearly collapses a bag to a fixed affine combination (e.g., sum/mean, or fixed-position linear weights) can realize the XOR configuration after pooling, which is not linearly separable. No downstream linear head can resolve this.

## B.3 Second impossibility: collision under convex-combination pooling

**Lemma 1** (Convex-hull collision)**.** *If two bags $X, Y$ satisfy $\mathrm{co}(X) \cap \mathrm{co}(Y) \neq \emptyset$, then any pooling that outputs a convex combination $\phi(\cdot) \in \mathrm{co}(\cdot)$ admits $X, Y$ with $\phi(X) = \phi(Y)$.*

*Proof.* Let $z \in \mathrm{co}(X) \cap \mathrm{co}(Y)$. Then $z = \sum_i \alpha_i x_i = \sum_j \beta_j y_j$ with $\alpha, \beta \geq 0$ and $\sum_i \alpha_i = \sum_j \beta_j = 1$. Set the pooling weights to $a_i(X) = \alpha_i$ and $a_j(Y) = \beta_j$; then $\phi(X) = z = \phi(Y)$. $\square$

**Remark 1.** *Softmax attention (with any score function) satisfies $a_i(X) \geq 0$, $\sum_i a_i(X) = 1$, hence always outputs $\phi(X) \in \mathrm{co}(X)$. Lemma 1 implies that whenever two bags have intersecting convex hulls, there exist valid weightings that make their pooled outputs identical. If such a collision occurs during training, no linear head can separate those bags.*

**Concrete 2D example (mean/attention).** Let $X_+ = \{(1, 0), (0, 1)\}$ and $X_- = \{(0, 0), (1, 1)\}$. Then

$$\mathrm{co}(X_+) = \{(\alpha, 1 - \alpha) : \alpha \in [0, 1]\}, \quad \mathrm{co}(X_-) = \{(\beta, \beta) : \beta \in [0, 1]\},$$

which intersect at $(\frac{1}{2}, \frac{1}{2})$. Mean pooling yields $\phi(X_+) = \phi(X_-) = (\frac{1}{2}, \frac{1}{2})$. Thus even a single positive and negative bag can *collapse* to the same pooled vector under linear convex-combination pooling.

## B.4 What linear heads cannot fix

Suppose $\phi(X_+) = \phi(X_-)$. Then for any linear head $h(z) = Wz + c$,

$$h(\phi(X_+)) = W\phi(X_+) + c = W\phi(X_-) + c = h(\phi(X_-)).$$

Therefore, no linear post-processing can undo a collision created by linear pooling.

## B.5 Geometric condition for linear pooling to succeed

A necessary condition for linear separability after convex-combination pooling is that the sets of pooled vectors be linearly separable. Because $\phi(X) \in \mathrm{co}(X)$, this requires the family of convex hulls of positive bags to be strictly linearly separable from those of negative bags. In many MIL problems—including pathology WSIs—positives and negatives often share large regions of feature space (e.g., stromal/normal contexts), making convex-hull intersections common and separability fragile.

## B.6 CONNECTION TO OUR METHOD

Our approach keeps the encoder/aggregator weights frozen but *learns the input `[CLS]` token* inside a transformer. Unlike linear convex-combination pooling, the `[CLS]` token participates in *nonlinear*, depth-wise self-attention:

$$\text{Attn}(Q_c, K, V) = \text{softmax}\left(\frac{Q_c K^\top}{\sqrt{d_k}}\right) V,$$

where $Q_c$ depends on the learned token. Across layers, this induces a nonlinear map of the entire bag that can separate XOR-type configurations and avoid convex-hull collisions by reshaping which instances contribute and how they interact. Empirically, this yields the consistent gains we report in Section 5.

