# OpenReview forum: "CLS-Tuned Attention for Whole-Slide Image Classification"
_ICLR.cc/2026/Conference — ICLR 2026 Conference Withdrawn Submission_

### Official Review · Reviewer_px6e · 2025-10-26

**Soundness:** 2
**Presentation:** 1
**Contribution:** 2
**Rating:** 2
**Confidence:** 5

**Summary:**

This paper proposes to fine-tune a CLS token in attention-based multi-instance learning (MIL) for WSI classification applications. Compared to existing MIL methods, it only fine-tunes the CLS token in MIL while keeping the rest of the MIL network frozen. Based on three WSI datasets, the authors show that the proposed method often obtains performance gains over the counterparts (those without CLS-token tuning). Moreover, the authors visualize the bag-level representations to intuitively show the effectiveness of the proposed method.

**Strengths:**

- This paper presents a new angle to improve MIL for WSI classification, i.e., fine-tuning the CLS token while keeping the rest frozen. It differs from most existing works in the field.
- The results on three public datasets show promising performances of the proposed CLS-tuned technique.

**Weaknesses:**

- Low presentation quality: This paper seems to be a hasty work. Its current form of presentation and organization may not be able to meet the standard of ICLR. For example, wrong uses of cite and citep, sudden and confusing texts at lines 78-83, incoherent texts (read more like drafts) at lines 108-137, and so on.
- Overclaims: The authors are encouraged to make the claims of this paper rigorous enough. For example, statistically significant accuracy gains at line 62 yet statistical tests are not conducted, faster convergence at line 63 but no evidence is presented, and so on.
- Weak theoretical contribution: In the theoretical analysis, the authors seem to fail to justify the proposed CLS-tuned method directly. Instead, that part focuses more on justifying the separability of linear and non-linear systems.
- Most datasets that are commonly used in WSI classification benchmarking, such as TCGA-BRCA, TCGA-RCC, etc, are missing in the experiments. The authors are encouraged to conduct experiments on these commonly-adopted WSI datasets.
- TransMIL also uses the CLS token in MIL. Could the author discuss the difference?

**Questions:**

Please see Weaknesses

---

### Official Review · Reviewer_VfoT · 2025-10-31

**Soundness:** 3
**Presentation:** 2
**Contribution:** 2
**Rating:** 4
**Confidence:** 4

**Summary:**

This paper introduces CLS-Tuned Attention MIL (CT-MIL), a parameter-efficient strategy for whole-slide image (WSI) classification. Instead of fine-tuning entire aggregators or encoders, CT-MIL adapts only a learnable [CLS] token and an optional lightweight CLS-gate, while keeping backbone weights frozen. This design leverages [CLS] as a global query to nonlinearly reweight patch embeddings, improving slide-level predictions without altering interpretability. The authors provide theoretical justification, showing that linear pooling often fails to ensure separability, whereas CLS-tuned attention restores it. Experiments on multiple histopathology datasets demonstrate consistent accuracy gains (+4–6%) across diverse backbones, with fewer parameters and faster convergence. CT-MIL is drop-in, architecture-agnostic, and deployment-friendly, offering a practical alternative to full fine-tuning for large-scale pathology applications.

**Strengths:**

1. The motivation is reasonable, since standard MIL fine-tuning is parameter-heavy, unstable, and prone to overfitting, so a lightweight alternative is needed. The idea of tuning only the [CLS] token for MIL in whole-slide image (WSI) classification is a simple but underexplored angle.

2. The paper evaluates CT-MIL across multiple datasets (EBRAINS, TCGA-NSCLC, CPTAC), using several MIL backbones (ABMIL, TransMIL, DSMIL, etc.), and reports balanced accuracy and F1 scores to account for class imbalance.

3. CT-MIL reduces the computational and data burden of WSI model adaptation, making it feasible for small-to-medium cohorts and clinical settings with limited resources.

**Weaknesses:**

1. The novelty of the method is limited. The central idea that tuning only the [CLS] token within an attention-based MIL aggregator is conceptually simple and can be seen as a parameter-efficient variant of existing fine-tuning strategies. While effective, this approach may appear incremental rather than a fundamentally new architectural design, since the [CLS] token has long been used as a global readout in transformer models.

2. The paper compares CT-MIL only against full fine-tuning and frozen-feature baselines. It does not benchmark against other parameter-efficient tuning methods (e.g., adapters, LoRA, or prompt tuning), which are also designed to work with frozen backbones.Limited comparison to other parameter-efficient tuning methods. The paper compares CT-MIL only against full fine-tuning and frozen-feature baselines. It does not benchmark against other parameter-efficient tuning methods (e.g., adapters, LoRA), which are also designed to work with frozen backbones.

3. The experimental section does not include cross-validation or repeated experiments with different seeds, and the mean and variance are not provided, making it difficult to effectively assess the robustness of CT-MIL.

**Questions:**

Please respond to the weakness with emphasis on the following points.

1. How does CT-MIL compare to other parameter-efficient fine-tuning methods?

2. The CLS-gate is introduced as an optional calibration tool, but its individual contribution is not fully isolated. Is the performance gain primarily from tuning the [CLS] token, or is the gate essential?

3. The paper claims preserved interpretability but only provides t-SNE plots at the bag level. The clinically relevant interpretability comes from patch-level attention maps. Are these maps more accurate or clinically plausible with CT-MIL?

---

### Official Review · Reviewer_An6W · 2025-11-01

**Soundness:** 3
**Presentation:** 3
**Contribution:** 2
**Rating:** 2
**Confidence:** 5

**Summary:**

This paper proposes CLS-Tuned Attention (CT-MIL), a simple yet effective framework for whole-slide image (WSI) classification. Instead of fine-tuning the entire attention-based MIL model, CT-MIL freezes both the patch encoder and aggregator, and only tunes a learnable [CLS] token plus a small CLS-gate for attention calibration. This design efficiently adapts global representations with minimal parameters and computation. Theoretical analysis shows that traditional linear pooling can fail to produce linearly separable bag representations, while a learned [CLS]-driven attention restores separability. Experiments on multiple pathology benchmarks (EBRAINS, TCGA, CPTAC) demonstrate consistent accuracy and F1 improvements across various MIL backbones.

**Strengths:**

(1). Simplicity and Practicality
The method makes only a minimal architectural change by tuning the [CLS] token but still achieves clear performance gains. It is simple to apply, requires no retraining of the encoder or aggregator, and works smoothly with existing MIL models.

(2). Parameter and Compute Efficiency
By freezing most of the network and updating only a few parameters, CT-MIL reduces training time, memory use, and overfitting risk, which is especially useful in medical imaging where labeled data are scarce.

(3). Comprehensive Empirical Validation
CT-MIL is tested across several datasets and backbone architectures, consistently improving accuracy and F1. The experiments include both internal and external validation, showing the model’s robustness and generalization ability.

**Weaknesses:**

(1) Limited Novelty and Overlap with Existing Work
The claimed contribution of introducing a CLS-tuned mechanism lacks sufficient novelty. Previous transformer-based MIL methods have already discussed the problem of redundant attention and introduced global or class tokens similar to the proposed [CLS] design, such as TransMIL [1] and Prototypical MIL [2]. Furthermore, several prototype-based or global representation MIL approaches, including TPMIL [3] and DGR-MIL [4], share very similar conceptual foundations. The idea of using a learnable global token to aggregate instance information has already been explored, making the innovation here incremental rather than fundamentally new. In addition, Attention-Challenging MIL [5] also examines how to refine attention in transformer-based MIL, which overlaps with this work’s motivation.



(2) Lack of Localization and Interpretability Analysis
In whole-slide image analysis, one of the main purposes of MIL is not only classification but also localization, particularly identifying tumor or abnormal regions within a slide. However, the paper does not provide a comparison or visualization of attention maps, especially between the [CLS] token and other patch tokens. There is also no analysis of patch-to-patch attention or Grad-CAM–like visualization to verify whether the model effectively captures local discriminative regions. This omission weakens the claim that the method improves interpretability.


(3) Unclear Treatment of Multi-class MIL Formulation
MIL is traditionally designed for binary classification, where a bag is labeled positive if it contains at least one positive instance. The datasets used in this paper are multi-class, yet the authors do not clarify how the MIL formulation is extended. If a one-vs-rest binary strategy was applied, it should be explicitly stated. If instead a standard cross-entropy loss was used, the method functions more like a conventional supervised classifier rather than a genuine MIL framework.



Ref:

[1]. TransMIL: Transformer based Correlated Multiple Instance Learning for Whole Slide Image Classification

[2] Yu, J. G., Wu, Z., Ming, Y., Deng, S., Li, Y., Ou, C., ... & Wang, Y. (2023). Prototypical multiple instance learning for predicting lymph node metastasis of breast cancer from whole-slide pathological images. Medical Image Analysis, 102748.

[3] Yang, L., Mehta, D., Liu, S., Mahapatra, D., Di Ieva, A., & Ge, Z. TPMIL: Trainable Prototype Enhanced Multiple Instance Learning for Whole Slide Image Classification. In Medical Imaging with Deep Learning. 2023

[4]. Zhu, W., Chen, X., Qiu, P., Sotiras, A., Razi, A., & Wang, Y. (2024, September). Dgr-mil: Exploring diverse global representation in multiple instance learning for whole slide image classification. In European conference on computer vision (pp. 333-351). Cham: Springer Nature Switzerland.

[5]. Yunlong Zhang, Honglin Li, Yunxuan Sun, Sunyi Zheng, Chenglu Zhu, andLin Yang. 2024. Attention-challenging multiple instance learning for whole slide image classification. In European Conference on Computer Vision. Springer, 125–143.

**Questions:**

See weakness above

---

### Official Review · Reviewer_qwa4 · 2025-11-01

**Soundness:** 2
**Presentation:** 2
**Contribution:** 2
**Rating:** 2
**Confidence:** 4

**Summary:**

This paper proposes a parameter-efficient multiple instance learning (MIL) approach for whole-slide image (WSI) classification, termed CLS-Tuned Attention (CT-MIL). The method fine-tunes only the Transformer’s [CLS] token and a lightweight CLS-gate, while keeping both the feature extractor and the MIL aggregator completely frozen. The authors further provide a theoretical justification showing that linear bag pooling may lead to non-separable representations, whereas a learnable [CLS] token can restore separability through nonlinear attention mappings. Experiments on several WSI benchmarks (EBRAINS, TCGA, CPTAC) show promising results. Overall, the paper presents an interesting idea, but the experimental validation, comparison with related methods, and writing quality are not sufficient to support acceptance.

**Strengths:**

1.Novel perspective. Introducing [CLS]-token tuning into MIL offers a new angle for parameter-efficient adaptation in computational pathology.
2.Practical simplicity. The method requires no architectural modification and can be easily integrated into existing MIL pipelines.
3.Initial empirical gains. The approach consistently improves over baselines across several MIL architectures.

**Weaknesses:**

1.Limited generalization study. All experiments use UNI features, without testing on other widely used WSI feature extractors (e.g., CTransPath, Virchow2).
2.Missing comparisons with Parameter-Efficient Fine-Tuning methods. The proposed approach essentially functions as a lightweight tuning mechanism built upon pretrained feature extractors and pretrained MIL aggregators. However, the paper does not provide systematic comparisons with mainstream Parameter-Efficient Fine-Tuning (PEFT) techniques such as Adapter, LoRA, or Prompt-based tuning. In addition, the claimed advantages in computational efficiency and training stability are not empirically substantiated — the paper lacks quantitative evaluations of parameter count, GPU memory consumption, training time, and convergence stability across different tuning strategies. Including such analyses would be crucial to validate the claimed benefits of CT-MIL over existing PEFT baselines.
3.No ablation for core components. The contributions of the learnable [CLS] query token and the CLS-gate (per-head vs. global) are not disentangled or analyzed separately.
4.Lack of a method diagram. A visual overview illustrating which parts are frozen and which are trainable would significantly improve clarity.
5.Outdated experimental scope. The experimental comparison is somewhat outdated, as it does not include more recent state-of-the-art MIL methods developed between 2023 and 2025.
6.Writing and organization issues. The paper contains redundancy errors, and the related work section lists many studies without clearly distinguishing this paper’s novelty.

**Questions:**

1.Can the authors provide experiments using different pretrained ViT feature extractors (e.g., CTransPath, Virchow2) to demonstrate generalizability?
2.How does CT-MIL compare to Adapter-, LoRA-, and Prompt-tuning methods in terms of accuracy, parameter count, memory usage, and training stability?
3.Could the authors include ablation studies isolating the effects of the learnable [CLS] token and CLS-gate (per-head vs. global)?
4.Would the authors add a method illustration to clarify the flow between frozen and trainable components?
5.Will the authors consider comparing to more recent MIL baselines such as ILRA-MIL (ICLR 2023) to better situate the proposed method?

---

### Note · Authors · 2025-11-12

I have read and agree with the venue's withdrawal policy on behalf of myself and my co-authors.